# Probing Electron Properties in ECR Plasmas Using X-ray Bremsstrahlung and Fluorescence Emission

Bharat Mishra [1,2,*], Angelo Pidatella [1], Alessio Galatà [3], Sandor Biri [4], Richard Rácz [4], Eugenia Naselli [1], Maria Mazzaglia [1], Giuseppe Torrisi [1] and David Mascali [1]

1. Istituto Nazionale di Fisica Nucleare—Laboratori Nazionali del Sud, via Santa Sofia 62, 95123 Catania, Italy; pidatella@lns.infn.it (A.P.); eugenia.naselli@lns.infn.it (E.N.); mazzaglia@lns.infn.it (M.M.); peppetorrisi@lns.infn.it (G.T.); davidmascali@lns.infn.it (D.M.)
2. Dipartimento di Fisica e Astronomia "Ettore Majorana", Università degli Studi di Catania, via Santa Sofia 64, 95123 Catania, Italy
3. Istituto Nazionale di Fisica Nucleare—Laboratori Nazionali di Legnaro, Viale dell'Università 2, 35020 Legnaro, Italy; alessio.galata@lnl.infn.it
4. Insitute for Nuclear Research (ATOMKI), Bem tér 18/C, H-4026 Debrecen, Hungary; biri@atomki.hu (S.B.); rracz@atomki.hu (R.R.)
* Correspondence: mishra@lns.infn.it

**Abstract:** A quantitative analysis of X-ray emission from an electron cyclotron resonance (ECR) plasma was performed to probe the spatial properties of electrons having energy for effective ionisation. A series of measurements were taken by INFN-LNS and ATOMKI, capturing spatially and spectrally resolved X-ray maps as well as volumetric emissions from argon plasma. Comparing the former with model generated maps (involving space-resolved phenomenological electron energy distribution function and geometrical efficiency calculated using ray-tracing Monte Carlo (MC) routine) furnished information on structural aspects of the plasma. Similarly, fitting a model composed of bremsstrahlung and fluorescence to the volumetric X-ray spectrum provided valuable insight into the density and temperature of confined and lost electrons. The latter can be fed back to existing electron kinetics models for simulating more relevant energies, consequently improving theoretical X-ray maps and establishing the method as an excellent indirect diagnostic tool for warm electrons, required for both fundamental and applied research in ECR plasmas.

**Keywords:** ECR plasmas; warm electrons; self-consistent simulations; experimental benchmarking; volumetric and space-resolved spectra; X-ray fluorescence; bremsstrahlung; ray-tracing Monte Carlo techniques

## 1. Introduction

Electron Cyclotron Resonance Ion Sources (ECRIS) are some of the most widespread devices used to generate highly charged ion beams of variable intensity to accelerators across the world. They are based on the dual concepts of ECR heating and magnetic confinement, whereby plasma electrons gyrating about a longitudinal magnetic field $B$ gain energy through resonance with circularly polarised electromagnetic (EM) radiation, and a min-B configuration traps them long enough to sequentially ionise atoms to high-charge states. The resultant plasma is composed of multi-charged ions immersed in a cloud of electrons of density $n_e \sim 10^{11}$–$10^{13}$ cm$^{-3}$ and temperature $k_B T_e \sim 0.1$–$100$ keV, which makes it ideal for not just ion beam generation, but also for research in other applied disciplines. For all its advantages, however, the system is quite difficult to study. The complex energy transfer process on account of the peculiar magnetostatic field profile and multi-modal nature of EM wave launched into the chamber [1], complicated transport phenomena [2,3], and the presence of instabilities [4] render the plasma non-homogeneous and anisotropic. Additionally, ECR plasmas are also known to support multiple electron

populations, of which the three most important classes are the cold ($k_B T_e \sim 10$–$100\,\text{eV}$), warm ($k_B T_e \sim 1$–$10\,\text{keV}$), and hot electrons ($k_B T_e \sim 10\,\text{keV}$-beyond).

Fundamental and application-oriented research require information on intermediate energy electrons spanning the boundary between warm and hot ($k_B T_e \sim 1$–$30\,\text{keV}$) because their properties govern the sequential ionisation process that forms the backbone of ECRIS operation. The PANDORA project is one such application which aims to measure $\beta$-decay rates of radioisotopes modified by the plasma environment, confining them in an ECR magnetic trap [5]. The idea is to then use these experimentally measured rates to verify the theory put forward by Takahashi and Yokoi [6] and in case of satisfactory match, extrapolate the same to stellar plasmas. The theory requires detailed inputs on the atomic level population and charge state distribution (CSD) of the ions which in case of ECR plasmas implies evaluating them as a function of position since the system is spatially non-homogeneous. The first step in achieving this is by characterising the spatial distribution of the aforementioned intermediate energy electrons.

To this effect, we present here a comprehensive analysis of the space-resolved properties of warm plasma electrons confined in compact magnetic traps, studied using a robust electron kinetics model and benchmarked with suitable experiments. In this article, we intend to elaborate on both aspects, but focus more on phenomenological plasma emission models connected with the latter, in an effort to demonstrate the utility of energy dispersive X-ray spectroscopy [7,8] when it comes to investigating intermediate energy electrons.

The contents of the article are divided into different sections as follows: in Section 2, the self-consistent electron kinetics simulations are briefly summarised and a quick overview is provided on determination of space-resolved electron energy distribution functions (EEDF). This effectively resulted in a theoretical 3D map of warm electrons whose properties were to be corroborated by experiments. Section 3 introduces the idea behind these validation experiments, and elaborates on the different kinds of measurements and their individual applicability. The focus is then shifted onto two particular experiments, namely 2D space-resolved X-ray imaging in photon counting mode and volumetric fluorescence/low energy bremsstrahlung emission spectroscopy, and corresponding theoretical models are presented, respectively, in Sections 4 and 5. By comparing the model-predicted results with those obtained experimentally, the two methods, respectively, furnished information about the spatial distribution of warm electrons, and estimates on the absolute number density of confined electrons and characteristics of escaping electrons. We conclude with Section 6, summarising the methodology employed and preliminary results on ECR plasma electron characterisation. The correctness of the latter is discussed with regards to the physics, and future improvements to the model are outlined in keeping with the requirements for the PANDORA project.

## 2. Space-Resolved Electron Kinetics: Theoretical Modelling

Numerical simulations are practical methods for probing microscopic properties of ECR plasmas because they offer reliable solutions to the several coupled differential equations describing such systems, which may otherwise remain unsolved. They are fast, accurate, and flexible with regards to model complexity. An iterative procedure to solve the collisional Vlasov–Boltzmann Equation [9] was developed to obtain a self consistent picture of stationary ECR plasmas. The simulations aimed to solve the plasma particles' equation of motion in the presence of an EM field self-consistently updated with the trajectories of the particles. Such an approach was essential because of the very nature of wave-particle interaction—the EM field dictates the motion of charged particles in the ECR plasma and the electrons' energy through resonant interactions, but the field profile itself is affected by the 3D dielectric tensor [10] calculated according to the spatial distribution of the particles, and hence to the electron density. The simulation model was generated using COMSOL Multiphysics® as an FEM solver and MATLAB® as a particle mover, based on a schematic outlined in Figure 1.

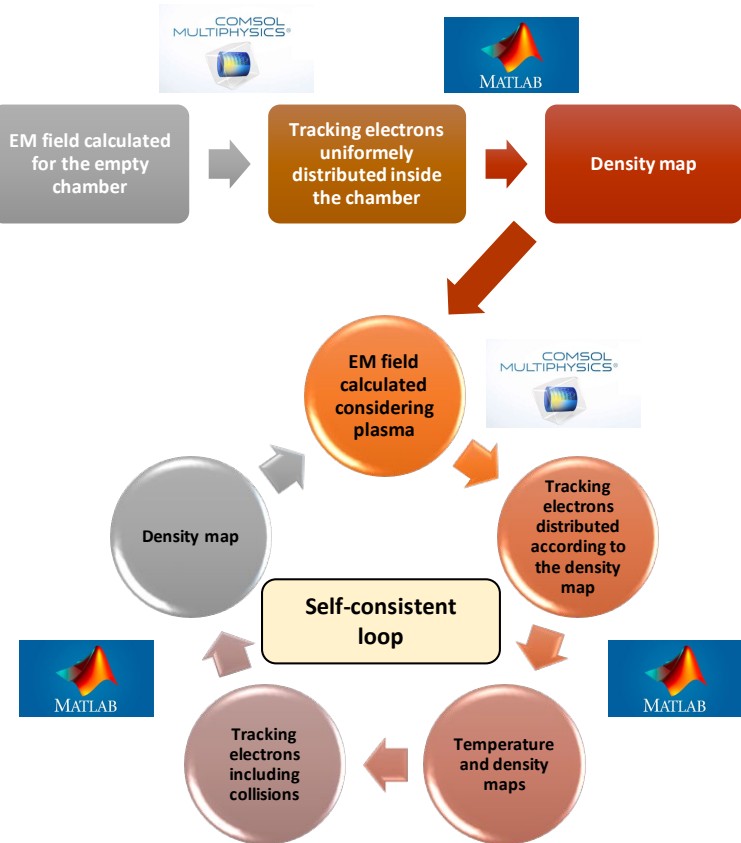

**Figure 1.** Schematic of self-consistent numerical simulations of warm electrons using COMSOL Multiphysics® and MATLAB®.

Details about the simulation scheme and associated algorithm can be found in [11], and experimental validation in [12–14]. The routine was first applied to warm electrons alone; the phase space trajectories of $N$ = 40,000 macroparticles were followed for a fixed simulation time $T_{sim}$ = 40 µs with time step $\tau_i$ = 1 ps and after each step, the code saved position and energy of particles in a 3D matrix corresponding to the simulation domain sliced into cells of 1 mm³, producing occupation and energy maps, respectively, [11]. The microwave frequency was taken as 12.84 GHz while the power was 30 W, matching the operating conditions of experiments as described in Section 3. At the end of the iteration routine (after achieving steady-state) the occupation maps were scaled to density maps by assuming a total number of particles spread out according to a plasmoid/halo structure [9], where plasmoid density ∼$10^{17}$ m$^{-3}$ and halo density one hundredth of it. The result was concise data on electron number and energy as a function of position in the plasma, expressed as a set of 7 number and 7 energy density matrices of dimensions 59 × 59 × 211, corresponding to the energy intervals most occupied by warm electrons [0, 2], [2, 4], [4, 6], [6, 8], [8, 10], [10, 12], and [12, ∞] keV. Some of the XY-projection maps (along the Z axis) are shown in Figure 2.

The discrete data thus obtained was subjected to post-processing for determining a phenomenological, space-resolved EEDF. First, the occupation matrices $\rho_i$ and energy density matrices $E_i$ were used to calculate average electron energy (AVE) in each cell according to the expression

$$\langle E \rangle = \frac{\sum_{i=1}^{7} \rho_i E_i}{\sum_{i=1}^{7} \rho_i}, \tag{1}$$

and the plasma was divided into finer *regions of interest* (ROIs) by grouping together cells with similar AVE. This facilitated analysis and helped resolving the inherent anisotropy

in the plasma. Figure 3 shows an isometric view of ROIs 1, 3 and 4, corresponding, respectively, to $\langle E \rangle$ = 0–0.1, $\langle E \rangle$ = 0.2–0.3, and $\langle E \rangle$ = 0.3–0.4 keV.

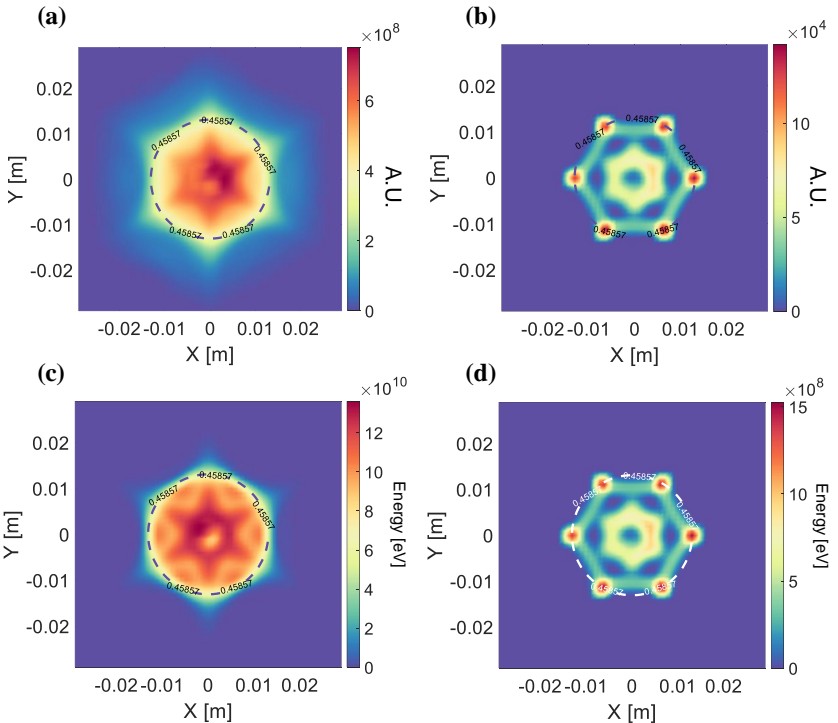

**Figure 2.** XY projection maps of electron density in (**a**) $[0, 2]$ keV, (**b**) in $[10, 12]$ keV, and energy in (**c**) $[0, 2]$ keV, (**d**) and in $[10, 12]$ keV, as resulting from self-consistent simulations at convergent stage, with $f = 12.84$ GHz and $P = 30$ W.

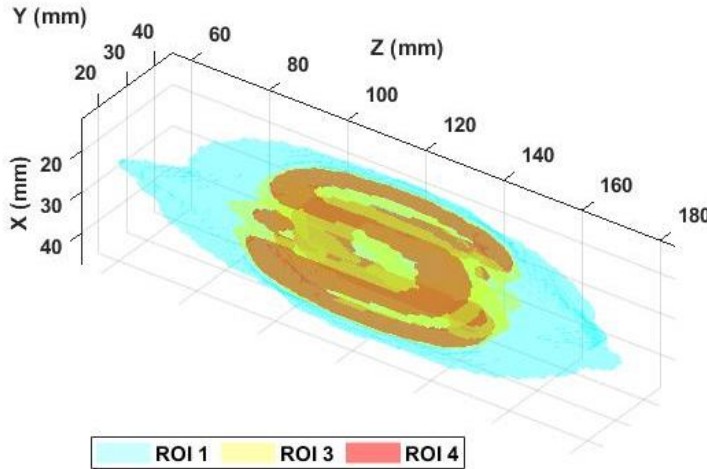

**Figure 3.** Isometric view of some AVE-based ROIs in the plasma.

The second step involved an exhaustive comparison of the performances of different EEDFs in the various ROIs based on statistical metrics like MSE and $r^2$, after which a two-component distribution composed of Maxwell and Druyvesteyn distribution as shown in Equation (2) was deduced as the phenomenologically and globally correct function.

$$f(E, k_B T_l, k_B T_h) = A_l \left( \frac{2}{\sqrt{\pi}} \frac{\sqrt{E}}{\sqrt{k_B T_l}^3} e^{-E/k_B T_l} \right) + A_h \left( 1.04 \frac{\sqrt{E}}{\sqrt{k_B T_h}^3} e^{-0.55E^2/k_B T_h^2} \right), \quad (2)$$

Here, $k_B T_l \sim 0.0086$–56.1 eV refers to the temperature of the Maxwell distribution function which described cold electrons accumulated in the $[0, 2]$ keV interval, while $k_B T_h \sim 1.1$–8.3 keV is the same for the Druyvesteyn function representing the warm electrons in the remaining energy intervals $[2, \infty]$ keV. $A_l$, $A_h$ are, respectively, the normalisation coefficients of the two components, while the energy intervals chosen could not sufficiently isolate the two populations, the phenomenological model adopted did manage to resolve them. It should be noted that though the form of the EEDF was seemingly uniform throughout the plasma, the defining parameters of the components, temperature and normalisation coefficients, varied spatially, reflecting the plasma non-homogeneity. More details on the choice of the EEDF, rationale behind multiple components, and statistical analysis can be found in [15,16]. Figure 4 shows a crude verification of the deduced EEDF where the aggregated number and energy density for a few ROIs are plotted against the same calculated using Equation (2)—the degree of match can be appreciated.

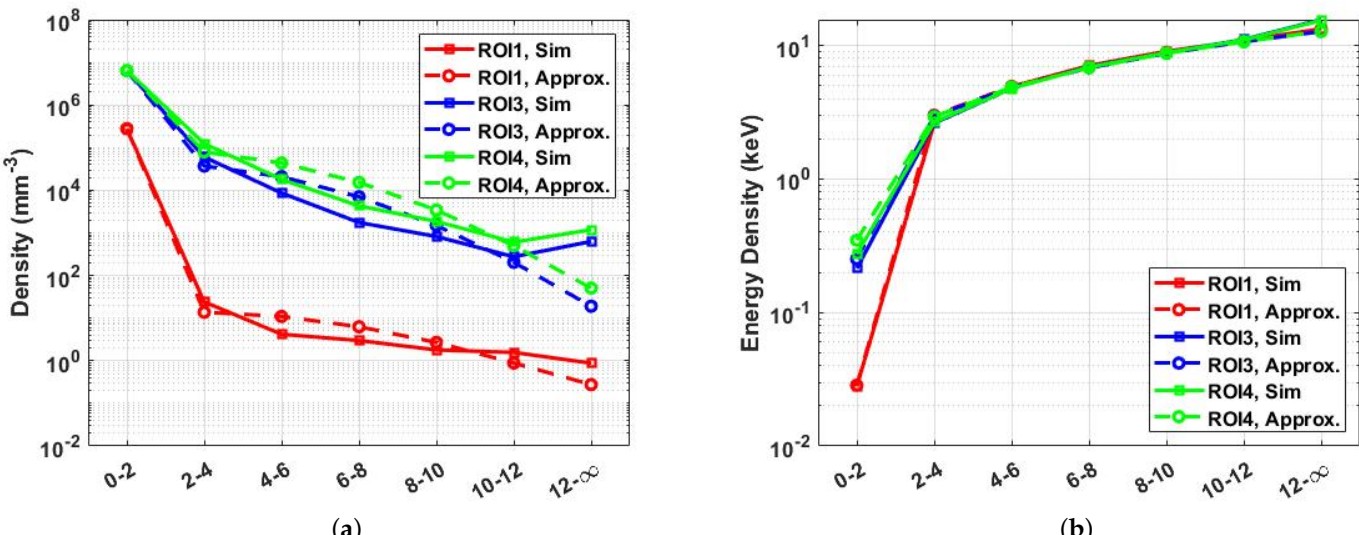

**Figure 4.** (**a**) EEDF estimated density vs. aggregated number density for ROIs 1, 3, and 4; (**b**) EEDF estimated energy density vs. aggregated energy density for ROIs 1, 3, and 4.

## 3. Energy Dispersive Soft X-ray Spectroscopy

ECR plasmas emit radiation across the EM spectrum, with each type of radiation reflecting the properties of some element of the plasma. As such, soft X-ray emission spectroscopy is a well-established tool for studying intermediate energy electrons [7,8,17–19] because the energy of the emitted photons lies in the range 2–30 keV. The spectra include both discrete fluorescence line emissions arising from the ionisation of the atoms, and continuous bremsstrahlung from the deceleration of electrons in the Coulomb field of the ions. Additionally, the experimental setup can be modified to study both global properties of the electrons (through volumetric spectroscopy) and local structure (through 2D space-resolved images).

The present work focuses on the analysis of soft X-rays from an Ar ECR plasma measured during an experiments at ATOMKI, Debrecen in 2014 [8], by developing a plasma emission model to explain the outputs from the silicon drift detector (SDD) and charge-coupled device (CCD) camera. The former was used for volumetric measurements alone, while the latter was intended for local structural studies by incorporating it into a more elaborate pinhole camera setup. The specific schematics for both setups are shown in Figure 5. The plasma chamber was of the same dimensions as the simulation domain, i.e., $59 \times 59 \times 211$ mm.

The SDD was coupled to a collimator that filtered out all photons outside of a narrow near-axis zone, allowing determination of plasma properties in a highly localised region.

The CCD-pinhole setup could be operated in two different ways. The first mode of operation was the spectrally integrated mode wherein the CCD chip was exposed to the plasma for tens of seconds and lost spectral resolution, but recorded the total photon energy impinging on it. This was useful for analysing the coarse shape and structure of the plasma as well as the local energy content.

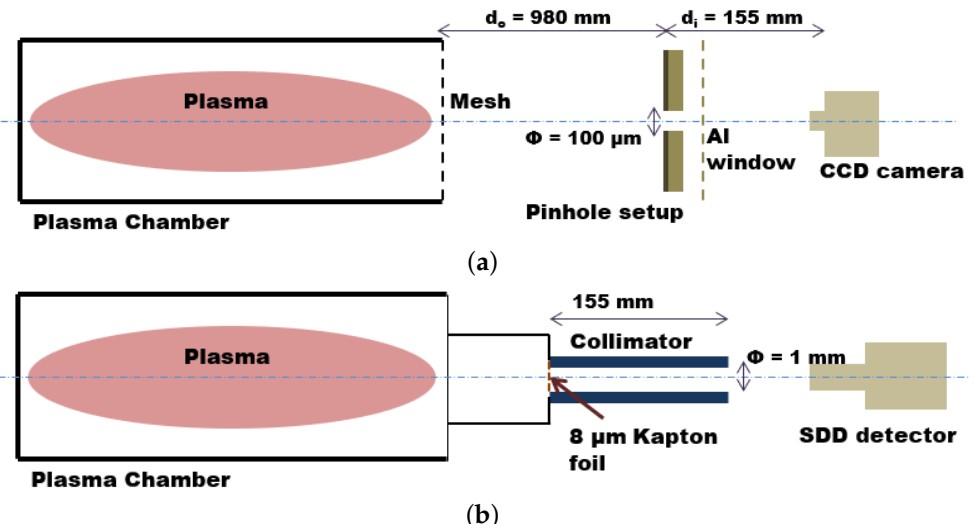

**Figure 5.** (**a**) Schematic of CCD-pinhole setup for 2D X-ray imaging; (**b**) schematic of SDD-collimator setup for narrow-zone volumetric spectrum measurement. The size of the plasma chamber coincided with that of the simulation domain.

The second mode of operation, and more relevant to present calculations, was the spectrally resolved or photon counting mode. Here, the CCD chip functioned as a fast camera, capturing a large number of ms-duration frames. This allowed each pixel to retain information about individual photons, resulting in a full spatially *and* spectrally resolved X-ray map showing the finer details about the plasma structure and local sources of fluorescence. More details about the aforementioned modes of operation and their importance in plasma studies can be found in [8].

## 4. The 2D Space-Resolved X-ray Imaging

In order to benchmark the theoretical model of warm electrons detailed in Section 2, the CCD-pinhole setup was operated in photon counting mode for an Ar plasma heated at operating frequency of 12.84 GHz and with RF power of 30 W. A total of 2000 frames were captured, each of duration 150 ms.

Experimental validation was attempted by comparing a theoretical fluorescence emission model based on the simulated $\rho_i$ map and Equation (2) with the Ar fluorescence map. The calculation was rather straightforward. The starting point was the volumetric reaction rate (in units $mm^{-3}s^{-1}$) defined as

$$R = \rho_e \rho_{Ar} \int_I^\infty \sigma(E) v(E) f(E, k_B T_l, k_B T_h) \mathrm{d}E, \tag{3}$$

where $\rho_e$ is the electron density, $\rho_{Ar}$ is the Ar ion density, $E$ is the collision energy in centre of mass (CM) frame of reference, $\sigma(E)$ is the K-shell ionisation cross-section as a function of collision energy, $v(E)$ is the collision speed, and $I$ is the K-shell binding energy (3.21 keV for Ar). The volumetric reaction rate could be converted into an emission map by calculating the measured photon counts from each cell (dimensionless units), given by the simple expression

$$T = 300 V \rho_e \rho_{Ar} Y \varepsilon_g \varepsilon_{h\nu} \int_I^\infty \sigma(E) v(E) f(E, k_B T_l, k_B T_h) \mathrm{d}E, \tag{4}$$

where 300 is the total exposure time in s, $Y$ is the fluorescence yield describing the percentage of K-shell ionisation converted into K$\alpha$ photons, $\varepsilon_g$ is the geometrical efficiency of the detector setup, and $\varepsilon_{h\nu}$ is the quantum efficiency of the CCD chip for detecting a 2.96 keV photon. $V$ is the volume of each cell which in this case was simply $1\,\text{mm}^3$. The electron density was nothing but $\rho_e = \sum_{i=1}^{7} \rho_i$.

While in principle Equation (4) fully described the spatially-resolved Ar K$\alpha$ emission map, there were some missing quantities and uncertainties. The ion density $\rho_{\text{Ar}}$ (and its spatial distribution) was unknown, but as a first approximation, it was taken as $0.25\rho_e$ assuming the plasma to be locally neutral and the ions to be in $4^+$ state. There was also uncertainty in the contribution of the warm electrons to the total emission, which could manifest as a mismatch between predicted and measured photon counts in the X-ray images. Furthermore, finally, due to the peculiar configuration of the imaging setup and the finite size of the plasma chamber, $\varepsilon_g$ was expected to be position-dependent as well, which entailed a more rigorous calculation of the spatially-resolved geometrical efficiency. These missing aspects will be addressed in the following subsections.

### 4.1. Assessing the Warm Electron Contribution—EEDF Integrated Cross Section

To understand better which electrons contribute most to the photon emission, the most straightforward way was calculation of the EEDF-averaged cross section defined as

$$\langle \sigma vs. \rangle = \int_I^\infty \sigma(E)v(E)f(E)\mathrm{d}E, \tag{5}$$

where $f(E)$ is the EEDF. Assessing the expected contribution of warm electrons from among the intermediate energy electrons is an important task—if hotter electrons with higher temperatures constitute the chief source of radiation, using only warm electrons in the theoretical emission model would eventually lead to underestimation of photon counts, or a corresponding overestimation in $\rho_e$ if theoretical and experimental maps were scaled to same order of magnitude. This analysis has already been detailed in [15], so only a brief overview will be provided here.

To evaluate Equation (5), the semi-empirical Lotz formula was used as the Ar ionisation cross Section [20] while the EEDF was taken as a pure Maxwell distribution. The upper limit of the integral in Equation (5) was truncated to 2000 keV for sake of brevity. Figure 6a shows an overlap plot of the Lotz cross section vs. Maxwell EEDFs with different temperatures, while Figure 6b shows the EEDF-integrated cross section. It can be easily concluded that the overlap between warm electrons as described in Section 2 and the cross section is weak as compared to slightly hotter electrons with $k_B T_e \sim 20$ keV and thus the emission map of Equation (4) will likely predict fewer photons than seen experimentally.

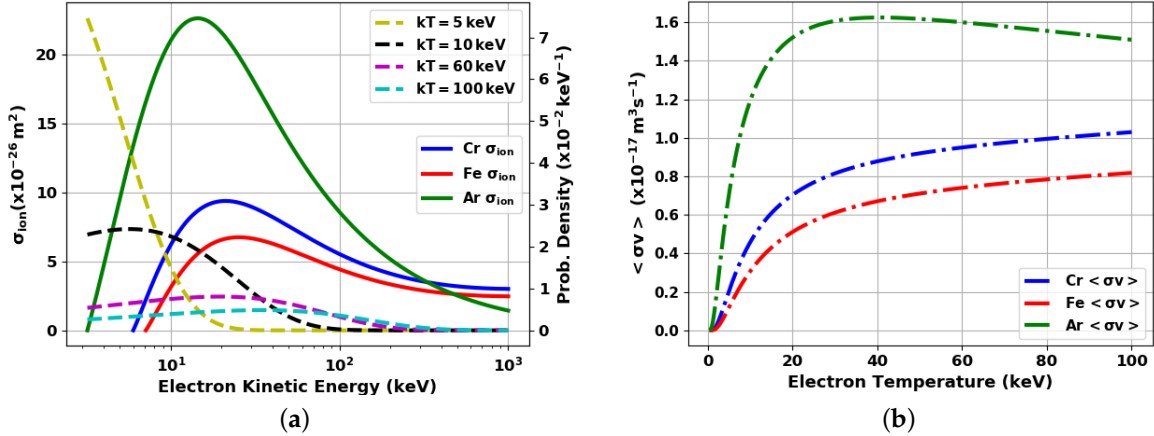

**Figure 6.** (**a**) Cross sections vs. Maxwell EEDFs of different $k_B T_e$; (**b**) EEDF averaged cross Section [15].

### 4.2. Evaluation of Local Geometrical Efficiency (LGE)

When dealing with isotropic radiation from a point-sized source, the geometrical efficiency of the setup is given by the simple expression

$$\varepsilon_g = \frac{\Delta\Omega}{4\pi}, \tag{6}$$

where $\Delta\Omega$ is the solid-angle subtended by the detector on the source and $4\pi$ is the full emission solid angle. This formalism cannot be adopted for the present case because the (1) the plasma is not one single point-sized source but rather a collection of point-sources of finite size, (2) the solid angle of emission is subtended by the *source* on the detector and not the other way round, and (3) each cell (point source) behaves differently with regards to $\varepsilon_g$ as defined by the optics of the cell and the pinhole setup. Figure 7 summarises the key ideas presented in these points, while imperceptible in the schematic, the emission space and detection probability of a photon varies spatially with the position of the source cell in the plasma.

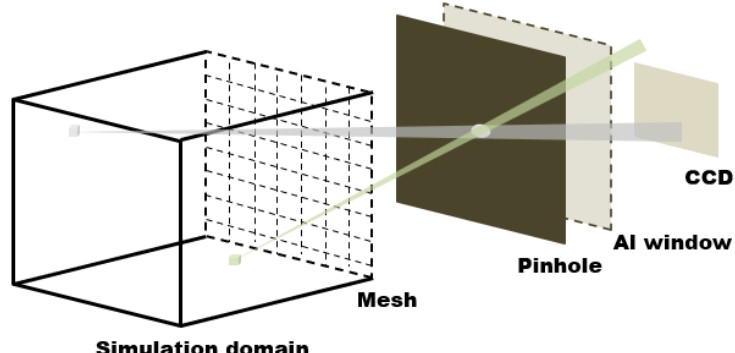

**Figure 7.** Schematic view of the plasma simulation domain, pinhole plane, Al absorbing window, and the CCD.

To address the aforementioned issues, a ray-tracing Monte Carlo method was used to calculate the geometrical efficiency of each cell in a cuboidal simulation domain of the same format as the 3D density and energy density matrices, thus called the *local geometrical efficiency* (LGE). The strategy was quite simple and based on a two-step approach. First, assuming each cell to be an independent source, the limits of the polar and azimuthal angles were calculated such that the photon when emitted within this defined emission space, would surely pass through the pinhole. Normally this would be an unnecessary step and one could straight away proceed with simulating a $N$ photons emitted isotropically from each cell and check how many passed through the pinhole. This is a computationally expensive procedure here since the pinhole blocks a majority of the photons emitted, and thus the resultant emission space is severely limited. Knowing the minimum and maximum of the polar/azimuthal angles, the emission space could be calculated as

$$\Delta\Omega = \int_{\theta_{min}}^{\theta_{max}} \int_{\phi_{min}}^{\phi_{max}} \sin\theta \mathrm{d}\phi \mathrm{d}\theta = (\cos\theta_{min} - \cos\theta_{max})(\phi_{max} - \phi_{min}), \tag{7}$$

and the *ideal* LGE would then be given by Equation (6). Figure 8a shows a rough sketch for calculating the emission space and Figure 8b shows a 3D sliced view of $\Delta\Omega$ for each plasma cell. The subtle variation across the simulation domain can be appreciated.

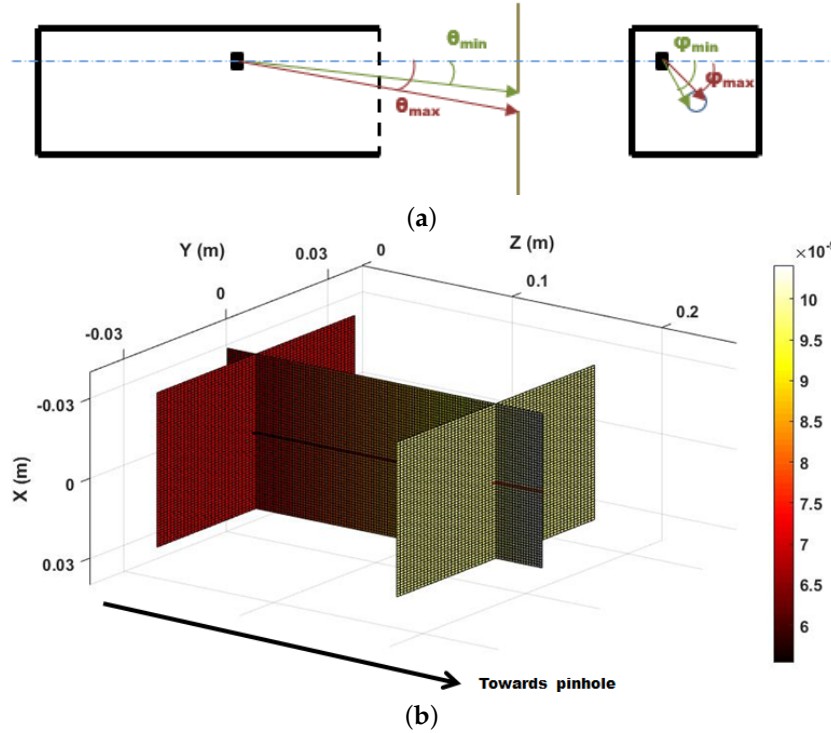

**(a)**

**(b)**

**Figure 8.** (**a**) Rough sketch demonstrating calculation of polar and azimuthal angles; (**b**) space-resolved solid angle in the plasma simulation domain.

However, simply passing through the pinhole is not a guarantee of detection—the photon could very well be hindered by some absorbing element or simply not intercepted by the CCD chip. To account for this, $N$ photons *within* the above calculated emission space were simulated, and their trajectories traced along their journey through the setup. The number of photons $N'$ which made it till the CCD were checked, and a correction $N'/N$ was factored into Equation (6) to obtain the final LGE. Figure 9a demonstrates the idea of ray-tracing while Figure 9b shows the final LGE. The setup of the experiment was modelled based on data provided in [18]—the 100 µm diameter pinhole was drilled into a stack of tungsten and lead plates of thickness 1 and 0.2 mm, respectively, and an aluminium window of thickness 3 µm was placed after the pinhole to block UV photons. Together with the mesh at the end of the plasma chamber, they acted as X-ray absorbers. The impact of the ray-tracing correction can be immediately confirmed by looking at the dark band of cells at both edges of the simulation domain—these are the cells from which emitted photons pass through the pinhole but never make it to the CCD. The lack of cylindrical symmetry is due to a combination of factors like assumed cuboidal shape of the simulation domain and limited resolution in terms of cell size. Furthermore, it is important to note the equalising effect of the absorbing elements on the LGE (since the final $\varepsilon_g$ looks more uniformly distributed as compared to Figure 8) and this is an artefact of the low statistics in the Monte Carlo routine. To retain the original distribution, a minimum of 10,000 photons should be simulated but currently only 500 were simulated. This does not have a large impact on the absolute results, however.

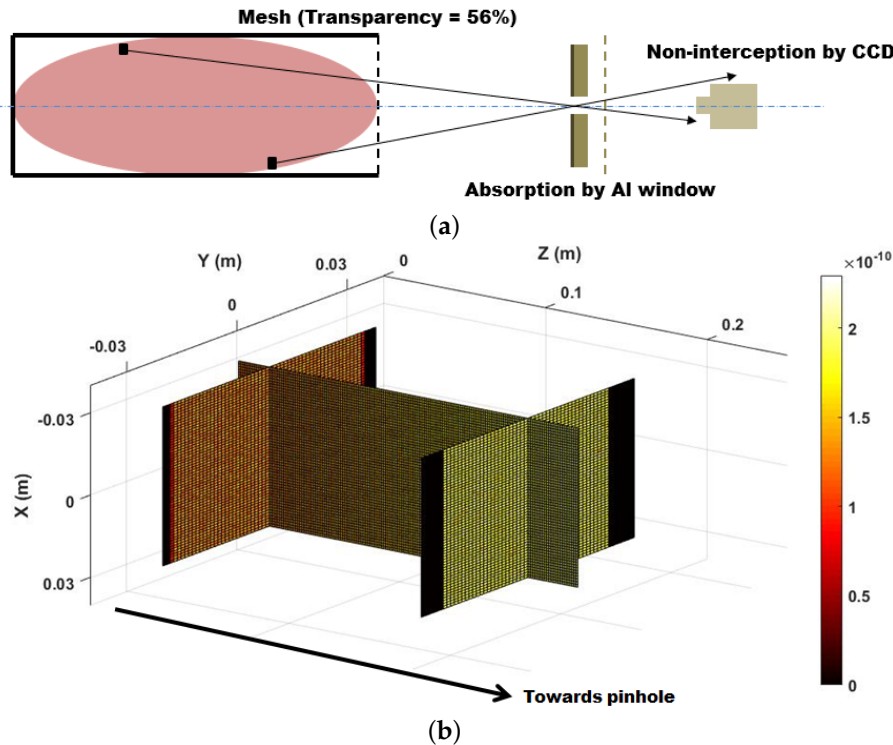

**Figure 9.** (**a**) Rough sketch demonstrating ray-tracing; (**b**) final $\varepsilon_g$.

### 4.3. Experimental Benchmarking

Using Equation (4), Lotz formula for Ar K-shell ionisation cross section, a fluorescence yield $Y = 0.119$, $\varepsilon_{hv} = 10\%$ and the calculated LGE, a theoretical emission map was obtained. The integral was evaluated with a Trapezoidal integration method and truncating the upper limit to 33 keV (the EEDF, and thus the integral, practically vanished after this limit). The global electron density was crudely re-scaled to $10^{18}$ m$^{-3}$, just so a match could be made with experimental data. Figure 10a shows the sliced emission map and Figure 10b shows the more relevant longitudinally integrated image which should be compared to the real map in Figure 10c.

Comparing the theoretical and experimental maps, it can be seen that the rough shape and structure of the plasma have been reproduced, as well as the "hole" in the near-axis region also seen in other experiments. There are of course differences, which can be attributed to a number of reasons like incorrect assumptions in ion distribution (they may not be locally neutral with the electrons [2]), negligence of photon scattering, and incomplete modelling of CCD readout. In addition, reconstruction of the number of counts came at the expense of re-scaling $\rho_e$ from $10^{17}$ to $10^{18}$ m$^{-3}$ which arises from evaluating the reaction rate using warm electrons alone. These issues, however, only seem to imply that our model is yet incomplete but does hold potential for studying spatial distribution of electrons in ECR plasmas. Corrections to the LGE evaluation scheme, improvement to ion dynamics simulation and a model of CCD action are still underway, while the problems associated with uncertainties in electron density/temperature will be addressed through volumetric spectroscopy.

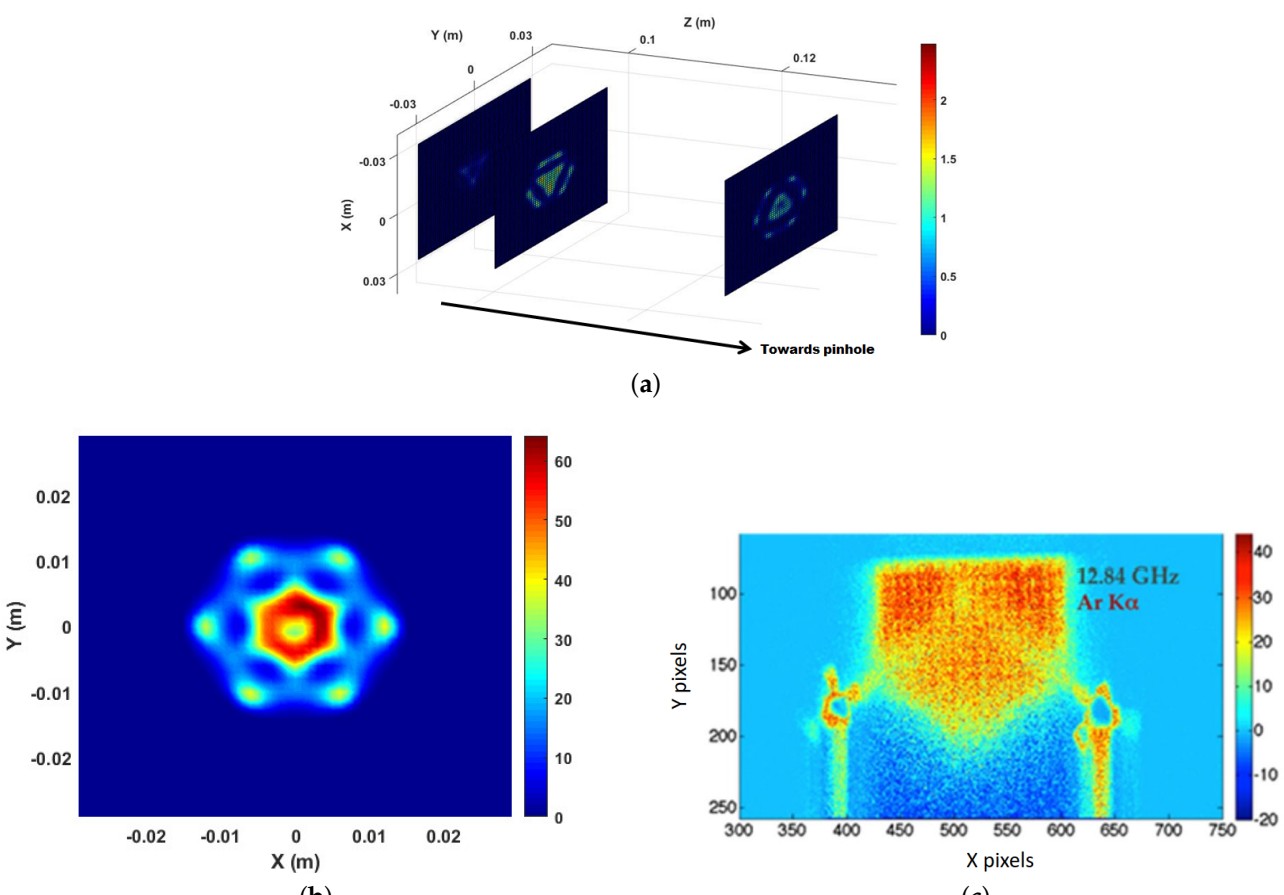

**Figure 10.** (**a**) Sliced Kα fluorescence map from Ar plasma, (**b**) longitudinally integrated, and (**c**) experimentally measured map. Reproduced with permission from Rácz, R. et al., Plasma Sources Sci. Technol.; published by IOP Science, 2017.

## 5. Volumetric X-ray Spectroscopy

To resolve the uncertainties about which electrons constitute the bulk of the fluorescence emission, analysis of the volumetric spectrum was done. The experiment was performed on the same Ar plasma heated with 12.84 GHz RF at 30 W power. The setup was as shown in Figure 5b and additional details about the measurement scheme/results can be found in [19]. Since many details about the the present analysis have already been published in [15,19], only a broad overview will be given here.

The bare spectrum from the SDD/ADC was first calibrated with Fe lines, corrected for quantum efficiency (QE) and dead time, and then converted to emissivity density $J_{h\nu}$. The presence of Kα and Kβ emissions from Ar, Cr and Fe was confirmed from the experimental emissivity density.

The emission model was built on the premise that a single component Maxwell distribution $f(E, k_B T_e)$ represented the entire emission zone. Comparison of model-predicted results with experimental data would validate the assumption. $J(h\nu)$ was composed of three different components—bremsstrahlung from confined plasma electrons, fluorescence from plasma ions (Ar lines), and fluorescence from extraction plate atoms generated by escaping electrons (Cr and Fe lines). The bremsstrahlung emissivity density $J_{theo,brem}(h\nu)$ equation was taken directly from [19], expressed as

$$J_{theo,brem}(h\nu) = \rho_e \rho_{Ar} (Z\hbar)^2 \left( \frac{4\alpha}{\sqrt{6m_e}} \right)^3 \left( \frac{\pi}{k_B T_e} \right)^{1/2} e^{(-h\nu/k_B T_e)}, \tag{8}$$

Here, $m_e$ is the electron mass, $\alpha$ is the fine structure constant, $h\nu$ is the photon energy, $Z = 18$ is the atomic number of Ar, $k_B T_e$ is the temperature parameter of the Maxwell EEDF, and $\rho_e \rho_{Ar}$ is the product of the electron and ion number density.

The Ar K$\alpha$ and K$\beta$ line emissivity densities were calculated according to the expression

$$J_{Ar,K\alpha(\beta)} = \frac{h\nu_{Ar,K\alpha(\beta)}}{\Delta E} \rho_e \rho_{Ar} \omega_{Ar,K\alpha(\beta)} \int_{I_{Ar}}^{\infty} \sigma_{Ar,ion}(E) v_e(E) f(E, k_B T_e) dE, \quad (9)$$

where $\omega_{Ar,K\alpha(\beta)}$ is the fluorescence factor associated with the transition, $\Delta E$ is the energy per channel, $I_{Ar}$ is the K-shell binding energy of Ar, and $\sigma_{Ar,ion}$ is the semi-empirical Lotz formula. The fluorescence factors are the same as fluorescence yield $Y$ in Equation (4) and $\omega_{Ar,K\alpha(\beta)}$ are connected to each other through the expression

$$\frac{I_{K\alpha}}{I_{K\beta}} = \frac{h\nu_{K\alpha}\omega_{K\alpha}}{h\nu_{K\beta}\omega_{K\beta}}, \quad (10)$$

where $I_{K\alpha}/I_{K\beta}$ is the line intensity ratio.

Finally, the emission from the extraction plate was calculated as

$$J_{X,K\alpha(\beta)} = \frac{h\nu_{X,K\alpha(\beta)}}{\Delta E V_P} \rho_{e,loss} n_X \omega_{X,K\alpha(\beta)} A \int_{I_X}^{\infty} v_e(E) f(E, k_B T_e) \int_{E}^{I_X} \frac{1}{S(E')} \sigma_{X,ion}(E') dE' dE, \quad (11)$$

where $X$ represents either Cr or Fe, $\rho_{e,loss}$ is the loss electron density, $V_P$ is the plasma volume introduced to convert the *total* extraction plate fluorescence emissivity density into a volume-averaged value, $n_X$ is the target atom number density, and $A$ is the area of the extraction plate interacting with the escaping electrons calculated as $A = 4\pi\varepsilon_g l^2 - \pi d^2/4$, $l$ being the separation between the extraction plate and detection cone vertex and $d$ the diameter of the extraction hole. This area was a a rather thin ring-like shape intercepted by the collimator on the extraction plate. Figure 11 shows the schematic for calculating the area. The quantity $S(E)$ represents the electron stopping power in AISI 316 steel which constituted the extraction plate, and was inserted to account for the constant modification of the ionising power of lost electrons as they traversed the target material.

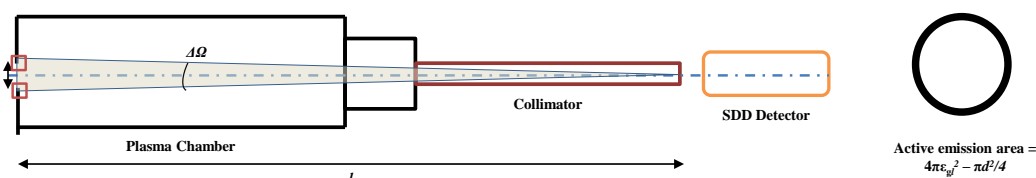

**Figure 11.** Area subtended by collimator on extraction plate for escaping electron fluorescence.

For $\sigma_{X,ion}$, the Lotz cross section was replaced with the Deutsch-Märk formalism [21]. Analogous to Section 4.1, the relative contribution of warm and hot electrons was also checked for Cr and Fe by plotting the overlap plot of Deutsch-Märk cross section with Maxwell EEDFs of different temperatures, as well as the EEDF-integrated values. This is also shown in Figure 6 and here too electron population with $k_B T_e \sim 20\,\text{keV}$ look to contribute more strongly. These results align well with calculations done in [19] on bremsstrahlung from Figure 12 alone, where electron temperature was estimated around 21 keV.

The fluorescence emissivity density from Equations (9) and (11) were modulated with a Gaussian profile to account for the line broadening and added to Equation (8) to generate the full model comprising of all three components. This was fit to the experimental spectrum using a Trust Region Selective least squares fitting routine, and parameters of interest, namely $\rho_e \rho_{Ar}$, $k_B T_e$ and $\rho_{e,loss}$ were estimated as $1.36 \times 10^{32}\,\text{m}^{-6}$, 22.18 keV and $10^{12}\,\text{m}^{-3}$, respectively. The result of the fit is shown in Figure 12.

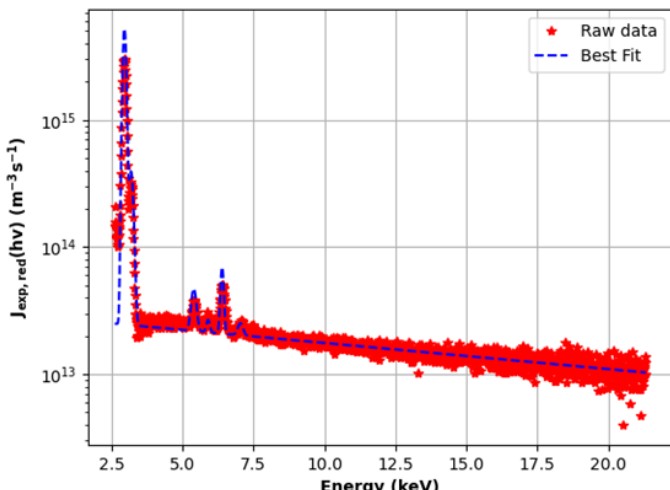

**Figure 12.** Final emissivity density marked with fluorescence lines of different elements in the plasma and the model fit.

The FWHM of the fluorescence peaks was estimated in the range 0.117–0.212 keV and the escaping electron current as calculated from $\rho_{e,loss}$ was around 2–5 mA/cm$^2$ which is the same order of magnitude as extracted ion current [19]. Just as importantly, the combined charge particle density $\rho_e \rho_{Ar} \sim 10^{32}$ m$^{-6}$ implies $\rho_e \sim 10^{16}$ m$^{-3}$ (if the ions are in 4$^+$ charge state) which is a fully valid result. This serves to prove that volumetric soft-X-ray spectroscopy is a powerful method to probe densities and temperatures of relevant electron populations and by coupling it to the 2D X-ray imaging analysis described in Section 4, the plasma can be completely characterised. For the moment, what it tells us is that warm electrons with in energy range 0–30 keV intervals are not sufficient to reproduce the degree of ionisation present in the plasma, and thus hotter species with $k_B T_e \sim 20$ keV should be simulated to match the experimentally obtained maps, both in terms of structure and absolute photon counts.

## 6. Conclusions

We have presented a detailed method for investigating the properties of intermediate energy electrons at the threshold between warm ($k_B T_e \sim 1$–10 keV) and hot ($k_B T_e > 10$ keV). A step-wise approach was implemented, starting with numerical simulations of warm electrons in energy range 0–30 keV and study of their spatial distribution through a deduction of phenomenological EEDFs that could effectively characterise them. Results of this theoretical model were subject to experimental verification using energy dispersive 2D X-ray fluorescence imaging and volumetric soft X-ray spectroscopy. By generating an emission model based on the theoretical electron maps and then comparing with the images experimentally captured using pinhole-CCD setup, the general shape and structure of the plasma was reproduced, but some differences remain. Most notable of the issues was the use of unphysically high electron density $\rho_e$ to match the photon count and uncertainty in degree of contribution from warm electrons. To quell these doubts, a volumetric emission model from the near-axis zone of the plasma was constructed and fit to experimentally measured soft X-ray spectrum from SDD-collimator setup. The results were a near-perfect fit using a basic, single component Maxwell EEDF of $k_B T_e \sim 22$ keV, with estimated $\rho_e \sim 10^{16}$ m$^{-3}$. This aligns with pure bremsstrahlung analyses made in [19] and predicts stronger contribution from hotter electrons to both bremsstrahlung and fluorescence. Using this information, we will update our electron simulations and populate higher energies, generate better emission maps and recheck match with 2D X-ray images. Simultaneously, we also plan to improve LGE evaluation by incorporating photon scattering effects, model the readout from the CCD chip and delve deeper into the distribution of ions. With continued efforts in terms of theory and experiment, energy dispersive soft X-ray spectroscopy can become a

handy technique to characterise electrons of importance in ECR plasmas and consequently improve our understanding of ion population kinetics for the PANDORA project.

**Author Contributions:** Conceptualization, S.B., R.R. and D.M.; Data curation, A.G., S.B., R.R. and D.M.; Formal analysis, B.M.; Funding acquisition, S.B., R.R., G.T. and D.M.; Investigation, S.B., R.R., E.N., M.M. and D.M.; Methodology, B.M. and A.P.; Project administration, G.T. and D.M.; Resources, A.G., S.B., R.R., E.N., M.M. and D.M.; Software, B.M., A.P., A.G., G.T. and D.M.; Supervision, A.P., A.G. and D.M.; Validation, E.N.; Visualization, B.M., A.P., A.G., E.N. and M.M.; Writing—original draft, B.M.; Writing—review and editing, A.P., A.G., S.B., R.R., E.N., M.M., G.T. and D.M. All authors have read and agreed to the published version of the manuscript.

**Funding:** This research received no external funding.

**Data Availability Statement:** The data that support the findings of this study are available from the corresponding author upon reasonable request.

**Acknowledgments:** The authors gratefully acknowledge the support of INFN by the Grant PANDORA_Gr3 (by the 3rd Nat. Comm.).

**Conflicts of Interest:** The authors declare no conflict of interest.

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
