# Peer review of "Probing Electron Properties in ECR Plasmas Using X-ray Bremsstrahlung and Fluorescence Emission"

_condensedmatter, doi:10.3390/condmat6040041_

Round 1

Reviewer 1 Report

Manuscript condensed matter-1425067

The manuscript presents a thorough analysis of the space-resolved properties of warm plasma electrons with magnetic confinement. It first provides a self-consistent electron kinetics simulation for the determination of electron energy distribution functions, which were then verified in experiments using energy dispersive 2D X-ray fluorescence imaging and volumetric soft X-ray spectroscopy. I find the paper well organized and adequately described, therefore I support the publication of this article in Condensed Matter. Here are a few minor questions and suggestions.

  1. For figure 3 and figure 4, what happens to ROI2 with <E>=0.1-0.2 keV?
  2. What is the size of the plasma chamber? The authors can add the dimension to Figure 5 (a). Is it the same as the simulation box of figure 3?
  3. Figure 10 (b,c) both axes need a label and unit
  4. Figure 12, why are the fitted emission peaks higher than the raw data?

Reviewer 2 Report

The paper looks good. It should be published.
Please check English again and amend some typos.

Detailed comments can be found at the attachment.
